# Rice Serine Hydroxymethyltransferases: Evolution, Subcellular Localization, Function and Perspectives

**DOI:** 10.3390/plants13081116

**Published:** 2024-04-16

**Authors:** Tian Pan, Hongmiao Jin, Chuanhui Zhou, Mengyuan Yan

**Affiliations:** The Key Laboratory for Quality Improvement of Agricultural Products of Zhejiang Province, College of Advanced Agricultural Sciences, Zhejiang A&F University, Hangzhou 311300, China; jhm@stu.zafu.edu.cn (H.J.); 202220020225@stu.zafu.edu.cn (C.Z.)

**Keywords:** serine hydroxymethyltransferase, rice, function

## Abstract

In rice, there is a lack of comprehensive research on the functional aspects of the members of the *serine hydroxymethyltransferase* (*SHMT*) gene family. This study provides a comprehensive investigation of the *SHMT* gene family, covering phylogeny, gene structure, promoter analysis, expression analysis, subcellular localization, and protein interaction. Remarkably, we discovered a specific gene loss event occurred in the chloroplast-localized group IIa SHMTs in monocotyledons. However, OsSHMT3, which originally classified within cytoplasmic-localized group Ib, was found to be situated within chloroplasts in rice protoplasts. All five OsSHMTs are capable of forming homodimers, with OsSHMT3 being the only one able to form dimers with other OsSHMTs, except for OsSHMT1. It is proposed that OsSHMT3 functions as a mobile protein, collaborating with other OsSHMT proteins. Furthermore, the results of *cis*-acting element prediction and expression analysis suggested that members of the *OsSHMT* family could be involved in diverse stress responses and hormone regulation. Our study aims to provide novel insights for the future exploration of SHMTs.

## 1. Introduction

One carbon metabolism is essential to all organisms. In plants, these reactions provide the necessary C1 units for protein synthesis, nucleic acids, pantothenate, and a diverse array of methylated compounds [1]. Despite the initial understanding of one carbon metabolism, many aspects of plant one carbon metabolism, including enzymes, pathways, and regulatory mechanisms, remain to be fully understood. Both serine and glycine serve as potential providers of C1 units in plants. The reciprocal transformation between serine and glycine is extensively studied in plant photorespiration and is integral in the synthesis of crucial compounds like purines, pyrimidines, and one-carbon units [1,2,3,4]. Serine hydroxymethyltransferase (SHMT, E.C. 2.1.2.1) is an important enzyme that catalyzes the conversion of serine and glycine and is widely distributed in plants, animals, and microorganisms [1]. SHMT facilitates the reversible transfer of a hydroxymethyl group from serine to H_4_PteGlu_n_ (tetrahydrofolate), resulting in the production of glycine and 5,10-CH_2_-H_4_PteGlu_n_ (5,10-methylenetetrahydrofolate) [5]. In plants, SHMT activity has been detected in different organelles (mitochondria, cytosol, plastids, and the nucleus), indicating their diverse roles in metabolic pathways [6,7]. Previous studies have shown that nearly all prokaryotic SHMTs exist in the form of homologous dimers, while eukaryotic SHMTs exist as homotetramers, which are dimers of dimeric structures [8]. It remains unclear whether SHMT proteins localized in different organelles within plants can form homologous or heterologous multimers.

Currently, the most extensively studied SHMTs are those localized within mitochondria. Earlier studies have found that mitochondrial serine hydroxymethyltransferase (mSHMT) has adapted to participate in photorespiration [9]. Within mitochondria, mSHMT and glycine decarboxylase (GDC) convert photorespiratory glycine into serine [4,10]. Arabidopsis mutants lacking mitochondrial SHMT activity exhibited lethal phenotypes when grown under ambient CO_2_ concentrations and are rescued under high CO_2_ [11,12]. AtSHMT1 enzyme activity plays a crucial role in controlling cell damage induced by high light and abiotic stress like salt [13]. The phosphorylation of SHMT1 at S31 might play a role in regulating SHMT1 protein stability, consequently regulating salt stress or drought stress in Arabidopsis [14]. The mitochondrially localized OsSHMT1 in rice is also a major protein influencing photorespiration. Studies have found that a mitochondrial *SHMT* gene mutation in rice (*osshmt1*) causes blockage of the photo-respiration pathway. *OsSHMT1* plays a role in reducing chloroplast ROS production to alleviate photoinhibition [15]. Additionally, Arabidopsis transgenic plants overexpressing *OsSHMT3* (actually *OsSHMT1*) exhibit salt stress tolerance, revealing the crucial role of *OsSHMT1* in enhancing plant tolerance to salt stress [16]. Recent research has reported the regulatory role of OsSHMT1 in rice under low-temperature stress, revealing its potential role in scavenging H_2_O_2_ to enhance cold tolerance in rice [17].

The presence of chloroplasts in plant cells makes one-carbon metabolism more intricate compared to animals and prokaryotes. Enzymes responsible for nucleotide biosynthesis reside within chloroplasts and the principal role of chloroplast SHMT is probably centered on utilizing serine within this space to supply a one-carbon unit [18]. Notably, there is a scarcity of studies focusing on SHMT proteins within chloroplasts. Although chloroplast-localized AtSHMT3 enzymatic activity was analyzed, it lacked corresponding mutant studies to further explore its impact on plant phenotypes and molecular mechanisms. The activity of SHMT in chloroplasts potentially plays a crucial role in light reception and the biosynthesis of purines, pyrimidines, and N-formylmethionine [7].

The one-carbon flux in the plant cytoplasm might predominantly engage in supporting SAM-mediated methylation reactions, as nucleotide synthesis seems not to occur within this compartment [19]. The products of cytosol, mitochondria, and plastid-located SHMTs can interconvert and communicate with each other [19]. Cytoplasmically localized GmSHMT08 in soybeans confers resistance to soybean cyst nematodes (SCN; *Heterodera glycines*) [20]. Recent experimental evidence indicates the involvement of GmSHMT08 in DNA methylation [21]. The GmSHMT08/GmSNAP18 (soluble NSF attachment protein, SNAP)/GmPR08-Bet VI (pathogenesis-related protein, PR) multi-protein complex is fundamental to soybean resistance against SCN [22,23]. Both THF and PLP sites at the GmSHMT08 are involved in resistance to SCN [24]. In conclusion, GmSHMT08 plays a pivotal role in the host defense process.

There is relatively limited research conducted on nuclear-localized SHMT. In Arabidopsis, AtSHMT7 in the cell nucleus regulates SAM biosynthesis and maintains the dynamic balance of sulfur via DNA methylation [6]. *OsSHMT4* negatively regulates the uptake and assimilation of sulfate/selenate, and its mutation enhances rice tolerance to Cd and selenium content in grains [25]. Furthermore, *OsSHMT4* influences SAM level and global DNA differential methylation, which further regulates the endosperm development [26].

In mammals, two SHMT isozymes are encoded by separate genes. *SHMT1* encodes the cytoplasmic/nuclear isozyme (SHMT1), and *SHMT2* encodes the mitochondrial (SHMT2) and the cytoplasmic/nuclear (SHMT2α) isoform through alternative promoter use [27]. The pathway for de novo thymidylate biosynthesis involves the translocation of enzymes like SHMT1 and SHMT2α to the nucleus during DNA replication and repair [27]. During the S and G2/M phases of the cell cycle, cytoplasmic SHMT1 has been demonstrated to relocate to the nucleus and nuclear periphery. This translocation is facilitated by post-translational modification through the attachment of the small ubiquitin-like modifier (SUMO) [28,29]. It remains unknown whether SHMT proteins from different organelles in plants undergo translocation. Although the SHMT family of proteins is relatively conserved, SHMTs with different signal peptides exhibit diversity in structure and function, participating extensively in the cellular life processes. The specific functions of SHMT proteins in different cellular locations within plants require further in-depth research.

In this study, a comprehensive analysis including the identification, evolution analysis, gene structure and promoter analysis, expression patterns in different tissues, responses to various hormones, subcellular localization analysis, and the interaction networks of SHMTs in rice was performed. These results contribute to the clarification of OsSHMT protein functions and provide valuable information for further studies in plants to uncover the functional roles of *SHMT* genes in growth and progression.

## 2. Results

### 2.1. Identification and Evolutionary Analysis of SHMT Family Genes in the Rice Genome

In this study, five *SHMT* genes were identified with a certain SHMT domain. The biophysical characteristics of these genes vary (Appendix A). The coding sequence length of the five *SHMT* genes was between 1416 and 1800 bp, and the predicted protein lengths of OsSHMTs were between 471 and 599 aa, with molecular weights (MWs) varying from 51.44 to 64.79 kDa. Among the five proteins, two have a pI smaller than 7, which were acidic proteins, while three OsSHMTs were projected to encode basic proteins with pI > 7. All OsSHMTs, excluding OsSHMT2, displayed an instability index (II) greater than 40, indicating a prevalence of unstable proteins within this gene family. It is worth noting that all OsSHMTs had a grand average of hydropathicity (GRAVY) of less than 0, indicating their hydrophilic properties. Among them, OsSHMT2 was the smallest protein of the five, with the lowest instability index and the biggest grand average of hydropathicity. All five SHMTs were distributed on different chromosomes but with similar MWs and hydrophilic characteristics.

To further evaluate the evolutionary relation among OsSHMTs, phylogenetic tree analysis was performed. The tree consists of 42 SHMT proteins, namely 7 in Arabidopsis, 12 in soybean, 8 in tomato, 10 in maize, and 5 in rice. According to the classification of Arabidopsis and soybean SHMTs. Plant SHMTs can be divided into four groups, group Ia, group Ib, group IIa, and group IIb (Figure 1). OsSHMT1 was grouped with five GmSHMT proteins (GmSHMT02m, GmSHMT08m, GmSHMT09m, GmSHMT14m, and GmSHMT18m) and two AtSHMT proteins (AtSHMT1 and AtSHMT2) in group IIb. OsSHMT2 and OsSHMT3 were grouped with AtSHMT4, AtSHMT5, GmSHMT05c, and GmSHMT08c and other five SHMT proteins in tomato, cucumber, and maize, classified into group Ib. OsSHMT4 and OsSHMT5 were placed in group Ia. Additionally, two AtSHMT proteins (AtSHMT6 and AtSHMT7), two CsSHMT proteins (CsSHMT5 and CsSHMT6), two ZmSHMT proteins (ZmSHMT3 and ZmSHMT6), four GmSHMT proteins (GmSHMT04n, GmSHMT06n, GmSHMT08n, and GmSHMT12n), and one SlSHMT1 protein were also included in this group. Notably, no homolog of AtSHMT3 was identified in rice (Figure 1), suggesting that a branch of the rice SHMT gene family is absent.

To verify whether the absence of group IIa SHMT in rice is also observed in other plant species, the extensive exploration of SHMT proteins among several representative species were carried out. SHMTs in *Populus trichocarpa*, *Nymphaea colorata*, *Amborella trichopoda*, *Chlamydomonas reinhardtii*, *Physcomitrium patens*, *Homo sapiens*, and *Saccharomyces cerevisiae* were added. The comprehensive analysis of the SHMT tree revealed a total of 79 SHMT proteins distributed among seven taxa, encompassing Eudicots, Monocots, Amborella, Chlorophyta, Bryophyta, Chordata, and Eumycophyta (Appendix A). The evolutionary tree is generally classification consistent with subcellular localization results, and all SHMT proteins can be categorized into two groups and four classes based on their cellular localization (chloroplastic, mitochondrial, cytosolic, and nuleaic) (Figure 2). The chloroplastic- and mitochondrial-localized SHMTs form one clade, while the cytosolic- and nuclear-localized SHMTs constitute another. Surprisingly, the chloroplastic class exclusively comprises proteins from dicotyledons, while the remaining subgroups encompass proteins from both monocotyledons and dicotyledons. Plant SHMTs likely underwent a duplication event, contrasting with their reduced numbers in humans and fungi. Within the same subgroup, the SHMT members of monocotyledons and dicotyledons clustered together, signifying a significant divergence in the evolution of SHMT genes between these two groups. Eudicots exhibit over three times the number of SHMT proteins compared to monocots, suggesting potential distinct evolutionary strategies adopted by dicot species.

### 2.2. Gene Structure and Promoter Cis-Element Analysis of OsSHMTs

To further understand the *OsSHMTs*, gene structure and conserved motifs were analyzed (Figure 3, Appendix A). The functional diversity of OsSHMTs is influenced by the unique compositions of the motifs. The analysis of conservative motifs revealed that five OsSHMTs shared motifs 1–7 and 9, with OsSHMT1 lacking motif 8, while OsSHMT4 and OsSHMT5 lacked motif 10 (Figure 3B). Previous studies have indicated that OsSHMT4 and OsSHMT5 individually lacked enzyme activity in vitro [26], suggesting that the absence of motif 10 could be pivotal for the enzyme activity of the SHMT family. The analysis of conserved domains demonstrated that all five OsSHMT proteins contain the SHMT domain, underscoring the conservative nature of their structure (Figure 3C). The divergence in exon–intron structure plays an important role in the evolution of duplicated genes. Then, we looked into the structure of the *OsSHMT* genes. The results showed that four *SHMT* genes contain four exons, but *OsSHMT1* has fifteen exons and all five *OsSHMT* genes have 3’ and 5’ UTR (Figure 3D). Although the majority of SHMT family members display some disparities in gene structure, substantial structural alterations have been noted in *OsSHMT1*, implying a differentiation mechanism during its evolution. The combined results of gene structure and phylogenetic trees indicate that OsSHMTs shared the SHMT conserved domain but underwent structural diversifications.

To elucidate the potential roles of *OsSHMT* family members in different responses and investigate their functions, a 2000 bp sequence upstream of the initial codons was extracted for a deeper understanding of the regulatory mechanisms of *OsSHMTs*. Our research revealed that in addition to light-responsive elements (such as G-Box, GT1-motif, and MRE), the promoter region of *OsSHMTs* contains a multitude of elements associated with growth, development, and responses to abiotic stress (Figure 4A, Appendix A). Growth and developmental elements include a meristematic tissue expression element (CAT-box and CCGTCC-box) and an endosperm expression element (RY-element and GCN4 motif). Hormone response elements encompass those that react to ABA (ABRE), methyl jasmonate (CGTCA-motif and TGACG-motif), auxin (TGA-element), gibberellin (GARE-motif and TATC-box), and salicylic acid (TCA). The stress response elements include wound response elements (WUN and WRE3 motif), dehydration response elements (DRE), drought response elements (MBS, Myb/Myc binding sites), stress-response element (STRE), anaerobic induction elements (ARE), and low-temperature response elements (LTR). These *cis*-elements found in the promoters of the *OsSHMT* genes have the ability to regulate the stress responsiveness or tissue-specific expression of the *OsSHMT* gene across various environmental conditions.

Apart from *OsSHMT5*, the promoters of the remaining *OsSHMT* family members harbor numerous ABA response elements. Additionally, the promoters of *OsSHMT2*, *OsSHMT3*, and *OsSHMT4* feature methyl jasmonate response elements. Furthermore, the promoters of *OsSHMT2* and *OsSHMT4* are rich in pressure response elements (Figure 4A). A large number of AP2/ERF family gene members can bind to the promoters of *OsSHMT2* and *OsSHMT4*, suggesting multiple AP2/ERF transcription factor binding sites within the promoter region. The MYB family was the only member predicted to be present in all five *OsSHMT* promoters (Figure 4B). These discoveries suggest that *OsSHMTs* may be influenced by various hormones involved in rice growth and its response to environmental stresses.

### 2.3. Expression Analysis of OsSHMTs in Various Tissues and in Response to Phytohormones

Considering that *cis*-acting elements are closely linked to gene expression, we first examined tissue-specific expression patterns for *OsSHMTs*. There is some variation in the expression patterns of homologous genes across different tissues. *OsSHMT1* was actively expressed in most young and mature tissues listed, especially in seeding roots and mature leaves. *OsSHMT2* only presented high expression in seeding roots, lateral inflorescence, and early seeds. *OsSHMT3* tends to be expressed in developed seeds and shares a similar pattern with OsSHMT4. The peak expression of *OsSHMT5* is shown in both young and mature leaves (Figure 5A). These results indicate that the expression of the *OsSHMTs* varies in different tissues to adapt to their functional differentiation.

As various phytohormone and stress responsive elements were distributed abundantly on *OsSHMT* promotors, we further analyzed the expression of *OsSHMTs* under different phytohormone treatments from 15 min to 6 h. The results showed that ABA treatment repressed the expression of *OsSHMTs*, while Jasmonic acid can induce the expression of *OsSHMT2*, *OsSHMT4*, and *OsSHMT5* and inhibit the expression of *OsSHMT3*, suggested that OsSHMT family proteins play an important role in biological and abiotic stress responses. *OsSHMT5* was suppressed by Cytokinin but induced by Auxin treatment, indicating that *OsSHMT5* may be involved in rice growth and development. Meanwhile, Cytokinin can increase the expression of *OsSHMT1* and *OsSHMT2* (Figure 5B). Treatment with different hormones triggers different changes in the expression of *OsSHMTs*, indicating that *OsSHMTs* can respond to multiple hormones, and there is significant functional differentiation among the homologous genes of *OsSHMTs*.

### 2.4. Subcellular Localization of OsSHMTs

Based on the evolutionary classification of OsSHMT family members, OsSHMT1 is localized in mitochondria, OsSHMT2 and OsSHMT3 are located in the cytoplasm, while OsSHMT4 and OsSHMT5 are located in the nucleus, and there are no homologous genes of SHMT localized in chloroplasts in rice. To verify the accuracy of the prediction results, we conducted subcellular localization analysis using rice protoplasts. We constructed the fused vector consisting of the coding sequence of *OsSHMTs* and GFP in the C-terminal driven by the cauliflower mosaic virus 35S (CaMV35S). Vectors to express the OsSHMTs-GFP fusion protein and empty-GFP (as a control) were introduced into rice protoplasts (Figure 6). Transiently expressed in the rice protoplasts, analysis demonstrated that OsSHMT1-GFP fusion protein presented both mitochondrial-like and cytosolic localization (Figure 6A), which is consistent with the previously reported results of the subcellular localization of OsSHMT1 [15]. Furthermore, OsSHMT2-GFP was localized in the cytosol of the transformed rice protoplasts (Figure 6B and Appendix A), whereas the OsSHMT4-GFP and OsSHMT5-GFP fusion proteins accumulated in the nucleus (Figure 6D,E and Appendix A), which is consistent with evolutionary classification. Remarkably, chloroplast-localized SHMTs are absent in monocotyledonous plants (Figure 2). Interestingly, OsSHMT3, classified within the cytoplasmic localization clade along with OsSHMT2, presented a chloroplastic-like localization (Figure 6C and Appendix A). Taken together, the rice SHMT protein family is localized in the mitochondria, plastids, cytoplasm, and nuclei, indicating their diverse roles in metabolic pathways.

### 2.5. Protein Interaction Networks of OsSHMTs

SHMTs are important enzymes that catalyze the conversion of serine and glycine. To deeply comprehend the potential function of OsSHMT proteins and the interaction between its family members, a protein–protein interaction network was predicted to explore the potential interacting proteins of OsSHMTs. There were five proteins predicted to interact with all the five OsSHMTs (Figure 7). According to the annotation, all five proteins have catalytic activities. Among them, LOC_Os01g51410, LOC_Os04g53230, and LOC_Os06g40940 participated in organic acid and amino acid metabolism. LOC_Os01g51410 and LOC_Os06g40940 were predicted to be glycine decarboxylase and can affect net photosynthetic rates as a key enzyme in the photorespiratory cycle [30,31]. Previous studies have shown that SHMTs in mitochondria cooperate with the glycine decarboxylase complex (GDC) to mediate photorespiratory glycine–serine interconversion in plants [10]. LOC_09g27420 was predicted to encode formate–tetrahydrofolate ligase, which is involved in the one-carbon pool by folate (Appendix A). Overall, proteins predicted to interact with OsSHMTs were annotated to be involved in carbon, glycine, and serine metabolism, which aligns with the overall functional role of OsSHMT proteins.

### 2.6. OsSHMTs Form Polymers

Studies have reported that the SHMT enzyme occurs as a tetramer in eukaryotes, because the active site is located at the interface of two monomers in an obligate homodimeric structure [32]. It is still uncertain whether SHMT proteins, localized in various organelles within plants, can form homologous or heterologous multimers. Based on the protein–protein interaction network, there is a potential for interaction among the homologous OsSHMT proteins. To further evaluate this possibility, we employed the yeast two-hybrid assay (Y2H) and found that all five OsSHMT proteins can interact with themselves (Figure 8A and Appendix A). Interestingly, OsSHMT3 can interact with other OsSHMT proteins, except OsSHMT1 (Figure 8B and Appendix A). In previous studies, we found that OsSHMT3 and OsSHMT4 can interact in the nucleus [26]. It seems that OsSHMT3 works in a similar way to SHMTs in animals, translocating in different organelles. OsSHMT3 may be the movable protein and cooperate with other OsSHMT proteins. Furthermore, OsSHMT4 and OsSHMT5 can also interact with each other.

The crystal structure and protein interaction model provide a way to learn how the OsSHMT complex was formed. The crystal structure of OsSHMTs were predicted by Alphafold. HDOCK and PyMOL were applied to imitate and visualize the model [33,34,35]. The result shows that OsSHMT3 forms tight-binding obligate dimers with OsSHMT2, OsSHMT3, OsSHMT4, and OsSHMT5 (Figure 9 and Appendix A). When OsSHMT3 interacts with OsSHMT2, the amino acids Ala75, Asp76, and Glu78 on OsSHMT3 form hydrogen bonds with Gln304 and Arg85 on OsSHMT2 (Figure 9A). Within the OsSHMT3 homodimer, a hydrogen bond is formed between Leu72 and Glu109 (Figure 9B). In the dimers of OsSHMT3 and OsSHMT4, the Thr175, Thr196, Thr210, and Try213 of OsSHMT3 are linked with Arg83 and Try86 of OsSHMT4 (Figure 9C). In the dimers of OsSHMT3 and OsSHMT5, the amino acids Asp76 and Glu78 on the former are linked with both Arg203 and Lys427 on OsSHMT5 (Figure 9D). The aforementioned findings indicate that the Leu72–Glu78 segment of OsSHMT3 plays an important role in the interaction between OsSHMT3 and its homologous proteins.

## 3. Discussion

### 3.1. The Functions of OsSHMTs Show Differentiation

Serine hydroxymethyltransferase is a crucial enzyme in the cell–carbon metabolic pathway, as it is responsible for the reversible interconversion of Ser and Gly [1]. Plants’ SHMT proteins can be classified into four groups according to their predicted subcellular localization [36]. The five *SHMT* genes in rice are categorized into three groups: group Ia, group Ib, and group IIb, with group IIa (chloroplast group) genes being absent. Remarkably, a specific gene loss event occurred in group IIa within monocotyledons (Figure 1 and Figure 2). The SHMT enzyme in plastids provides a carbon unit for the biosynthesis of 5,10-CH_2_-H_4_PteGlu_n_, which can then be oxidized by other catalysts into 5,10-CH^+^-H_4_PteGlu_n_ (5,10-methylene tetrahydrofolate) and 10-HCO-H_4_PteGlu_n_ (10-formyl tetrahydrofolate) [5]. The 5,10-CH^+^-H_4_PteGlu_n_ serves as a vital light-harvesting cofactor in plastid-localized cryptochrome [37]. Therefore, SHMT activity in plastids has a potentially important role in light reception. The abundance of SHMT proteins in plant species surpasses those found in humans and fungi, indicating a greater complexity in plant SHMT proteins and potentially novel functions (Appendix A). In monocotyledonous plants, alternative mechanisms may compensate for the absence of SHMT in group IIa. Interestingly, our subcellular localization analysis revealed that OsSHMT3 was located within chloroplasts in rice protoplasts (Figure 6), suggesting a gene duplication event within group Ib may compensated for this loss in group IIa. The aforementioned findings suggest that SHMT proteins in plants are more intricate and may possess novel functionalities. SHMTs usually function as homozygous or heterozygous complexes. The yeast two-hybrid assay results showed that all five OsSHMTs can form homodimers, and only OsSHMT3 can form dimers with other OsSHMTs, except OsSHMT1 (Figure 8). In silico docking simulations showed that at a certain angle, some residues can form hydron bonds shorter than 3 Å (Figure 9, Appendix A). In short, these results indicated that OsSHMT3 may serve as a mobilized protein and cooperate with other proteins. While the functions of SHMT1 and SHMT4 in rice have been relatively well elucidated, the understanding of other SHMTs is still limited. Previous studies have indicated that OsSHMT1 and OsSHMT3 have enzyme activity, whereas OsSHMT4 and OsSHMT5 individually lacked enzyme activity in vitro [16,26]. It is unclear whether these SHMTs primarily act as enzymes or serve purely regulatory roles. However, this hypothesis needs further investigation.

### 3.2. OsSHMTs Emerge as Promising Candidates for Bolstering Plant Resistance against Both Abiotic and Biotic Stressors

As stationary organisms, plants face a multitude of environmental stresses. Essential to their survival are plant hormones, which govern responses to both biotic and abiotic challenges. Notably, the ABA and JA signaling pathways, along with their interconnections, play pivotal roles in enhancing plant resilience against both biotic and abiotic stresses [38,39]. Environmental stresses induce changes in hormone biosynthesis, initiating the binding of transcription factors to specific *cis*-acting elements within promoter regions of downstream stress-responsive genes. This regulatory process governs gene expression, ultimately fine-tuning plant tolerance in reaction to abiotic stresses. The AP2/ERF and MYB transcription factor families are ubiquitous among plants, playing pivotal roles in regulating responses to abiotic stresses while also influencing plant growth and development [40,41]. In our results, the promoters of *OsSHMT* family members contain numerous ABA and JA response elements (Figure 4A). Moreover, *OsSHMT* promoters are rich in pressure response elements, along with numerous bound AP2/ERF and MYB transcription factors (Figure 4B). The analysis of promoters and the observed responsiveness of *SHMTs* to ABA and JA suggest that OsSHMTs are likely to play a crucial role in plant stress responses, serving as downstream targets for AP2/ERF and MYB.

Prior research has highlighted the significant role of Arabidopsis and rice SHMT1 in regulating cellular damage induced by abiotic stress [13,17]. Moreover, the GmSHMT08 gene in soybeans has been associated with imparting resistance against soybean cyst nematodes (SCN) [20]. OsSHMT4 negatively regulated the uptake and assimilation of mineral salt and improve rice resistance and grain quality [25]. Its homolog in wheat was found to be responsive to abiotic stress, abscisic acid, methyl jasmonate, hydrogen peroxide, and especially, Fusarium head blight [42]. *OsSHMT2* and *OsSHMT4* exhibit a significant abundance of *cis*-elements and transcription factor binding sites among the five *OsSHMT* genes (Figure 4). This suggests that *OsSHMT2* and *OsSHMT4* potentially hold greater significance in rice development. Hence, exploiting plant SHMTs to enhance crop resilience against both biological and abiotic stress represents a promising strategy. However, further investigation is required to uncover additional naturally occurring variations in *SHMT* genes associated with plant stress responses.

## 4. Materials and Methods

### 4.1. Identification and Phylogenetic Analysis of SHMT Family Genes in Rice Genome

The protein sequence of *Oryza sativa*, *Arabidopsis*, *Solanum lycopersicum*, *Zea mays*, and *Glysine Max* were downloaded from the phytozome database (https://phytozome-next.jgi.doe.gov/, accessed on 11 November 2023). The hidden Markov model (HMM) of the SHMT domain was obtained from Pfam (http://pfam-legacy.xfam.org/, accessed on 11 November 2023) with the ID of PF00464. All proteins were searched against the SHMT domain and then checked by NCBI (https://www.ncbi.nlm.nih.gov/, accessed on 11 November 2023). The properties of the OsSHMT proteins were predicted by Expasy (https://www.expasy.org/, accessed on 11 November 2023). Then, all the SHMT proteins were used to build the SHMT phylogenetic tree using MAFFT (version 7.380) [43], FastTree (version 2.1.7) [44], and MEGA 7 (version 7.0.9) [45] with default parameters. Proteins used to build the phylogenetic tree were listed in Appendix A. The tree was embellished by the online tool Evolview (https://www.evolgenius.info/evolview/#/, accessed on 25 November 2023) [46]. The aligned sequences of SHMT from different species were obtained from Gramene (https://plants.ensembl.org/Oryza_sativa/Info/Index, accessed on 5 January 2024), the tree was built by the maximal likelihood method using Fasttree and visualized by iTOL (https://itol.embl.de/, accessed on 19 January 2024)

### 4.2. Gene Structure Analysis of OsSHMTs

The gff file and genome sequence file of rice and corn were downloaded from phytozome (https://phytozome-next.jgi.doe.gov/, accessed on 11 November 2023). Protein and full-length coding sequences of *OsSHMTs* were analyzed by MEME (https://meme-suite.org/meme/, accessed on 25 November 2023) [47] and NCBI (https://www.ncbi.nlm.nih.gov/cdd, accessed on 25 November 2023), respectively. The visualization of the gene structure analysis was accomplished by TBtools-II (version 1.098667) [48].

### 4.3. Predicted Cis-Element and Transcription Factor Binding Sites on OsSHMT Promoters

The 2 kb upstream elements of the initiation codon were extracted as promoters. Then, Plantcare (http://bioinformatics.psb.ugent.be/webtools/plantcare/html/, accessed on 10 December 2023) [49] and PlantTFDB (http://planttfdb.gao-lab.org/prediction.php, accessed on 10 December 2023) [50] were used for *cis*-element and transcription factor binding sites, respectively, with default parameters. Excel and TBtools-II [48] were used to count and visualize the results.

### 4.4. Expression Analysis

The tissue-specific expression data were obtained from Rice Genome Annotation Project (http://rice.uga.edu/expression.shtml, accessed on 23 January 2024). The original expression data can be found in Appendix A. TBtools [48] was used to draw the heatmap with a log_10_(FPKM) value. The expression data of hormone treatment on 7-day-old seedling roots were downloaded from RiceXpro (https://ricexpro.dna.affrc.go.jp/data-set.html, accessed on 23 January 2024).

### 4.5. Protein–protein Interaction Prediction

The protein sequences of OsSHMTs were used for protein–protein interaction analysis by STRING (https://cn.string-db.org/, accessed on 25 January 2024) with default parameters. Gene annotation was also downloaded from STRING.

### 4.6. Subcellular Localization Analysis

The *OsSHMT* coding sequences were cloned into pAN580 vector with a GFP tag and driven by the double 35S (d35S) promoter. The primers used are listed in Appendix A. Rice seedlings were germinated and grown in half-strength MS liquid media within a growth chamber (14 h/30 °C light and 10 h/25 °C darkness) for 10 days before being used to generate protoplasts. Protoplast transformation was conducted following previously established procedures [51]. The OsSHMTs-GFP construct was introduced into the protoplasts, which were then incubated in darkness at 28 °C for 16 h prior to examination. Localization of the fusion protein used was determined under a confocal laser scanning microscope (LSM980 laser, Carl Zeiss, Oberkochen, Germany). GFP signals were recorded with the excitation wavelength at 488 nm and emission wavelength ranging from 505 to 530 nm, while mCherry signals were recorded with the excitation wavelength at 587 nm and emission wavelength ranging from 600 to 630 nm.

### 4.7. Yeast Two-Hybrid

The GAL4 system was selected for yeast two-hybridization. The coding regions of *OsSHMTs* were linked into pGBKT7 vector and pGADT7 vector, respectively. The constructed plasmids were co-transfected into yeast AH109 and cultured on SD/-Leu/-Trp solid medium and then transferred to SD/-Leu/-Trp/-His/-Ade solid medium to verify the interactions (the relevant vectors and reagents were obtained from Clontech). The primers used are listed in Appendix A.

### 4.8. Protein Structure Prediction

Protein sequences of OsSHMT were loaded on alphafold2.0 (https://colab.research.google.com/github/sokrypton/ColabFold/blob/main/AlphaFold2.ipynb, accessed on 5 February 2024) for the three-dimensional structure prediction. Proteins for molecular docking simulations were carried out by HDOCK (http://hdock.phys.hust.edu.cn/, accessed on 5 February 2024) and visualized by PyMOL (version 2.32).

## 5. Conclusions

In this study, a genome-wide analysis of the *SHMT* gene family in rice was conducted. We examined the physical and chemical characteristics, phylogenetic relationships, gene structure, motif scanning, promoter *cis*-elements, subcellular localization, and the protein interactions of the five SHMT proteins. Phylogenetic analysis revealed that the five OsSHMT proteins could be classified into three groups. Gene structure analysis showed that *OsSHMT1* lacks one motif but contains more introns compared to other *OsSHMTs*. The results from *cis*-acting element prediction and expression analysis suggested that *SHMT* family members may play various roles in multiple stress responses and hormone regulation. Subcellular localization analysis indicated that OsSHMT1–5 are localized in the mitochondria, cytoplasm, chloroplast, and nucleus, respectively. Protein interaction analysis demonstrated that OsSHMT3 can form both homodimers and heterodimers. The accumulation of variations during the evolution of OsSHMT led to neofunctionalization and subfunctionalization events. Our data will serve as a crucial resource for advancing research on the functional aspects of *OsSHMT* genes.

## Figures and Tables

**Figure 1 plants-13-01116-f001:**
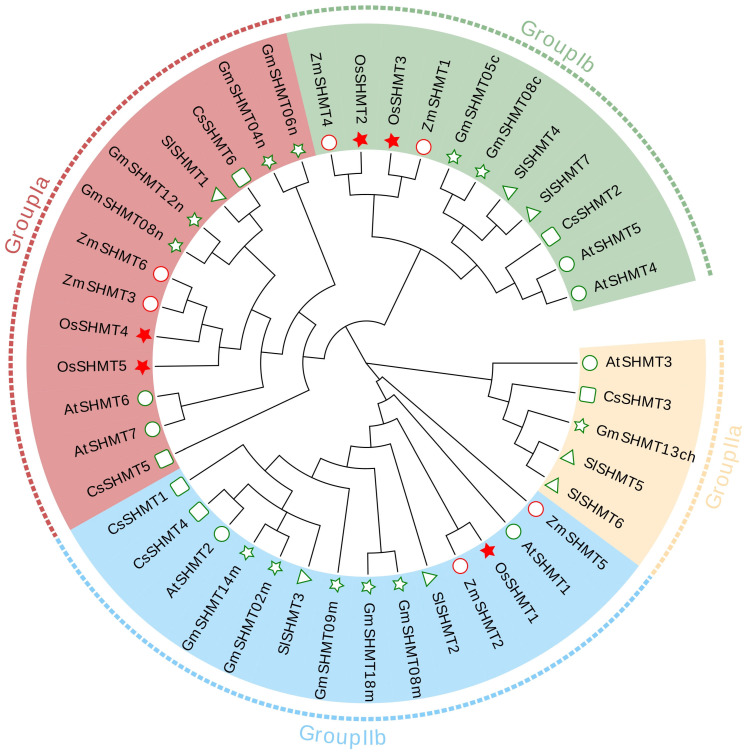
Phylogenetic tree of SHMT proteins from *O. sativa*, *Z. mays*, *A. thaliana*, *G. max*, *S. lycopersicum*, and *C. sativus*. The phylogenetic tree was built using the maximum likelihood method (ML) with 1000 bootstrap replicates by MEGA 7. Different colors represent different subfamilies, as follows: group Ia (red), group Ib (green), group IIa (orange), and group IIb (blue). Symbols with different colors represent the SHMT proteins from different species in this tree: *O. sativa* (red stars), *Z. mays* (red circles), *A. thaliana* (green circles), *G. max* (green stars), *S. lycopersicum* (green triangles), and *C. sativus* (green squares). Gene IDs of the analyzed genes can be found in Appendix A.

**Figure 2 plants-13-01116-f002:**
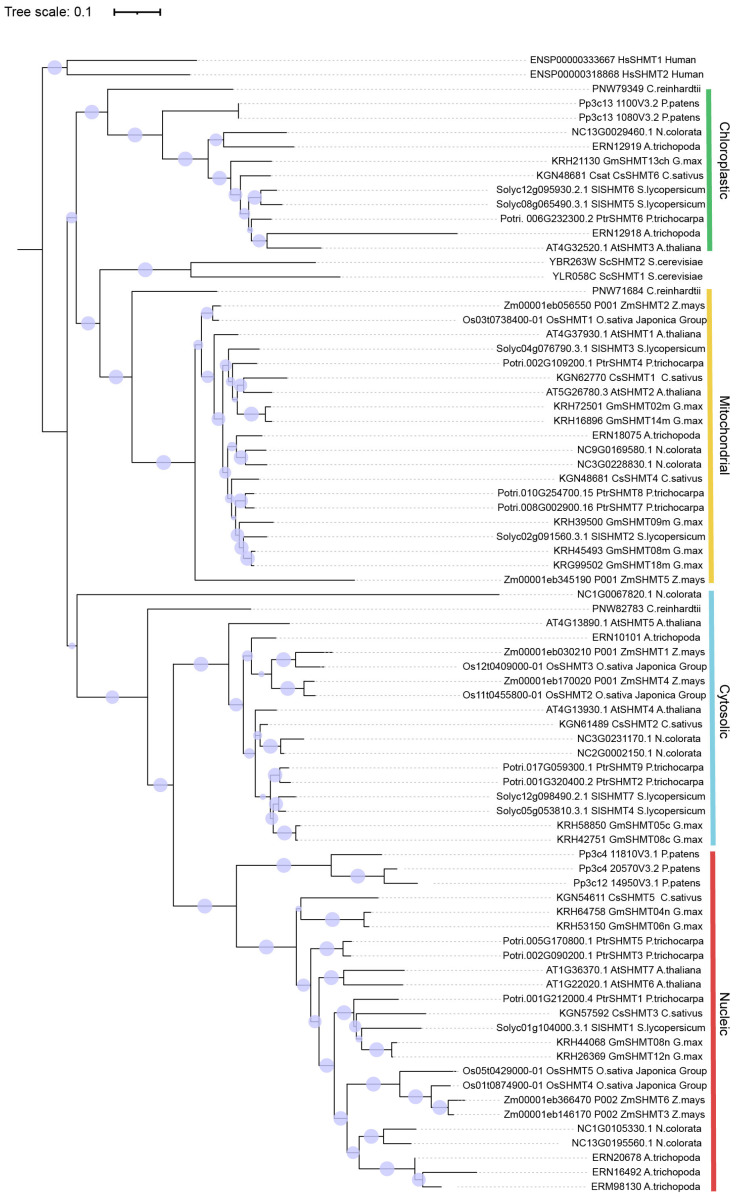
The evolution of SHMT proteins. The tree was constructed using trimmed amino acid sequences obtained from the ensemble (Appendix A). The phylogenetic tree was generated using MEGA7 and calculated using the maximum likelihood method. The tree bootstrap values are indicated at the nodes (*n* = 1000).

**Figure 3 plants-13-01116-f003:**
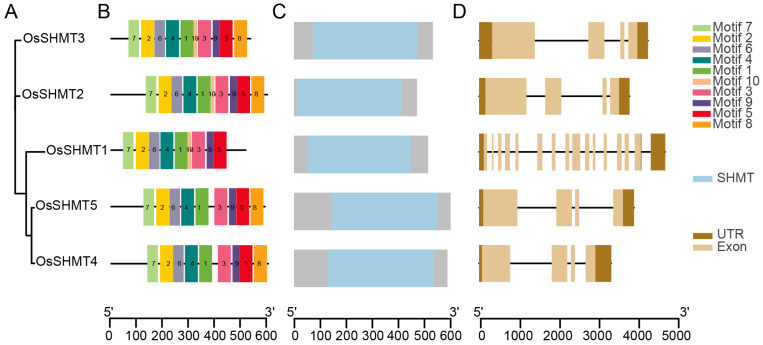
Phylogenetic relationships, conserved motifs, protein structure, and gene structure of *OsSHMTs*. (**A**) The phylogenetic tree was constructed using the maximum likelihood method with 1000 bootstrap replicates by MEGA 7. (**B**) The motif composition of SHMT proteins. The 10 motifs were indicated by colored boxes and numbered 1–10. (**C**) The conserved domain of OsSHMT proteins. Blue boxes for the SHMT domain. (**D**) The intron and exon distribution of *OsSHMT* genes. Light brown boxes indicate exons, dark brown boxes indicate untranslated sequence, and black lines indicate introns.

**Figure 4 plants-13-01116-f004:**
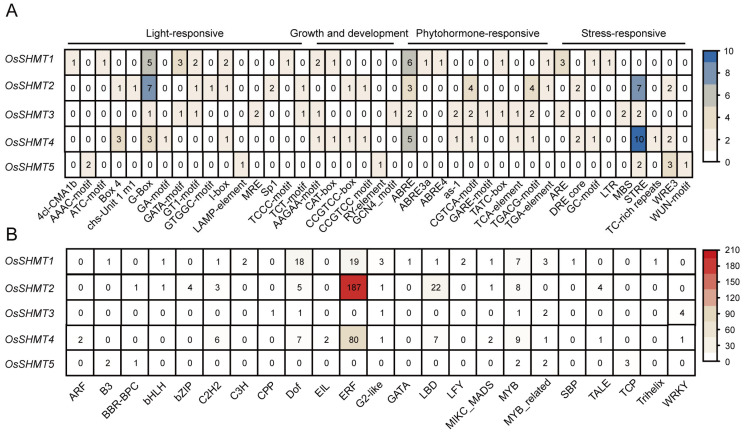
The prediction of *cis*-element and bound transcription factors on *OsSHMT* promoters. (**A**) The *cis*-elements distributed on the promoters of *OsSHMTs*. (**B**) Putative transcription factors binding to the *OsSHMT* promoters.

**Figure 5 plants-13-01116-f005:**
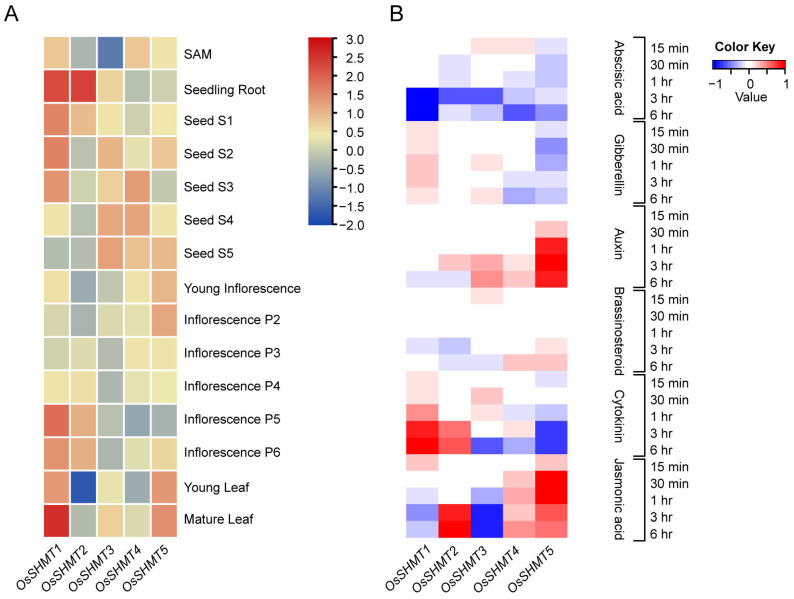
Expression analysis of *OsSHMT* genes during rice development and hormone treatment. (**A**) The expression profile of *OsSHMTs* in tissues. The samples were obtained from shoot apical meristems (SAM), seedling roots, five stages of developing seeds, six stages of developing panicles, young leaves, and mature leaves. The log10(FPKM) value was used to draw this heatmap. (**B**) The expression profile of *OsSHMT* genes under hormone treatment in rice seedling roots with a time course. The heatmap was downloaded from RiceXpro.

**Figure 6 plants-13-01116-f006:**
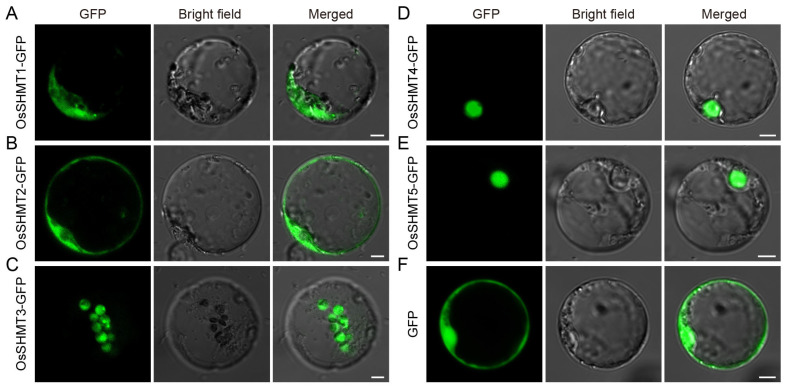
Subcellular localization of five OsSHMTs. (**A**–**F**) Confocal microscopic images showing the subcellular localization of five OsSHMTs in rice protoplast. GFP, fluorescence signals of the fusion proteins or free green fluorescent protein (GFP) proteins. Bars = 5 μm.

**Figure 7 plants-13-01116-f007:**
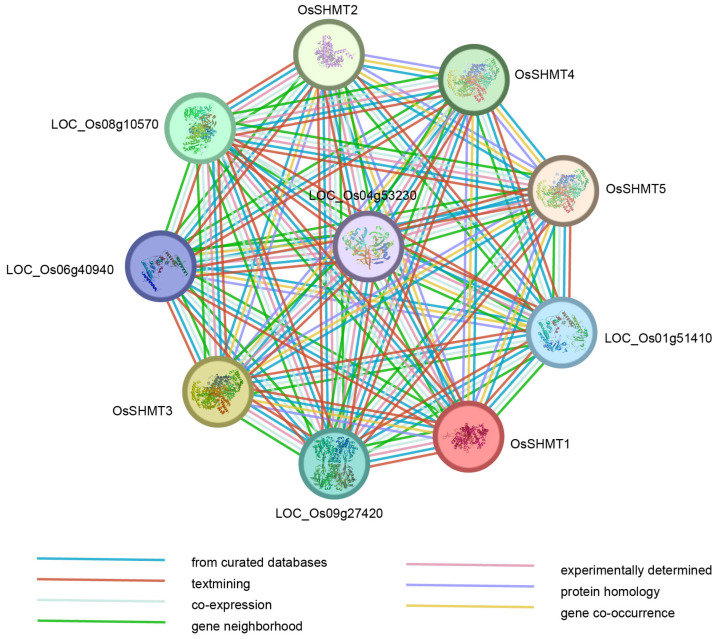
Protein interaction network of OsSHMTs. The nodes represent proteins, and the edges indicate the associations between two proteins. The types of evidence suggesting a functional link are distinguished by edges with different colors.

**Figure 8 plants-13-01116-f008:**
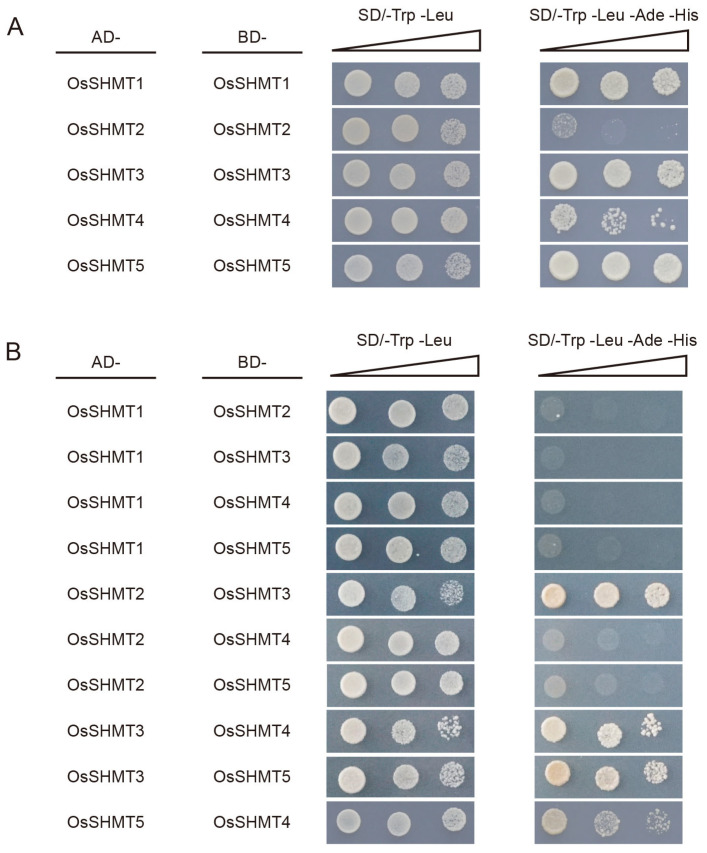
(**A**,**B**) Y2H assay shows that OsSHMTs interact with themselves (**A**) and their homologous proteins (**B**). DDO, SD/-Trp/-Leu; QDO, SD/-Trp/-Leu/-Ade/-His. AD, fused with activation domain; BD, fused with binding domain.

**Figure 9 plants-13-01116-f009:**
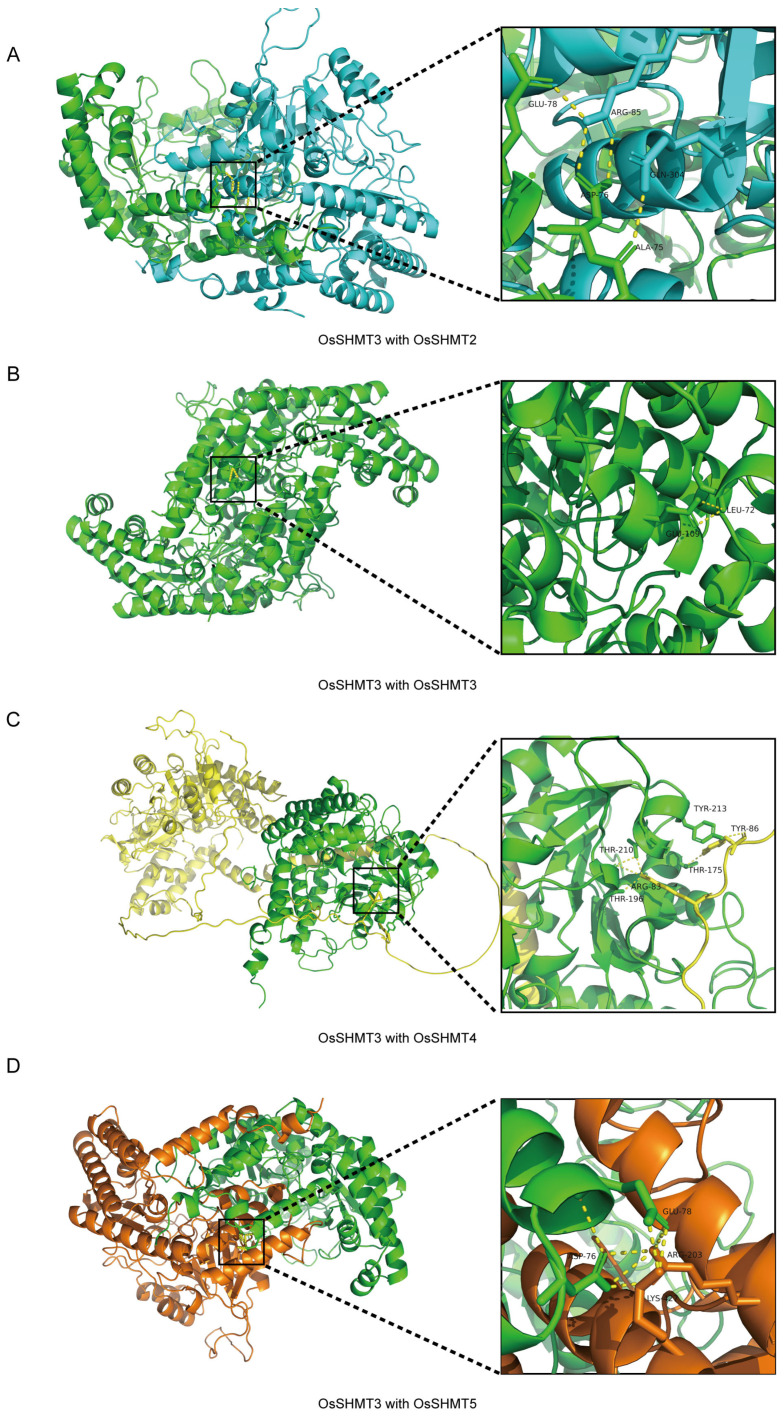
OsSHMT3 dimers and interfaces residues. (**A**–**D**) The interaction of OsSHMT3 with OsSHMT2 (**A**), OsSHMT3 (**B**), OsSHMT4 (**C**), OsSHMT5 (**D**). Differently colored proteins denoted different OsSHMT proteins. OsSHMT2 for blue, OsSHMT3 in green, OsSHMT4 in yellow, and OsSHMT5 in orange. Black squares show the zoomed interface of the dimers, and yellow lines represent the hydron bonds between interacting amino acids.

## Data Availability

The original contributions presented in the study are included in the article/Appendix A.

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
