# Peer review of "Rice Serine Hydroxymethyltransferases: Evolution, Subcellular Localization, Function and Perspectives"

_plants, 2024, doi:10.3390/plants13081116_

Round 1

Reviewer 1 Report

Comments and Suggestions for Authors

Authors studied the  serine hydroxymethyltransferases family in rice. Authors generally analysed the information available  in databases to perform the gene structure, protein sequences, promoter structure and evolution studies.

Also data related to organ-specific and hormone-responsive expression of these genes were provided. Moreover, the subcellular localization, interaction networks and Y2H experiments were performed.

Although the study is based generally on avialable data and a minor fraction (subcellular localization, protein structure or Y2H) is Authors own experimental work, the research provide still  interesting synthesis of data available for this gene family in rice and may be important to researchers in the field, particularly in part related to experimental Y2H protein-protein interactions.  Study is well planned and performed, results suport conclusions. Figures and tables are of good quality.

Particularly materials and methods section should be improved for better results reproducibility.

Following comments should be addressed to improve the manuscript:

Fig. 3

Provide sequences of presented protein motifs, together with their putative function.

 in the additional supplement file or table

Figure 4B- why so much trans-factors are bound by OsSHMT2 (187) and Os SHMT4 (80) ? Could Authors explain this difference based on own observations or references?

Section 4.6

Provide temperature, length of day and night, intensity of light-if available, during plant cultivation.

Provide name of applied vector. How the DNA fragments were inserted into vector-provide this information, method of cloning, for example applied restriction sites, how DNA fragments were amplified,

Provide details of protoplast preparing and  transformation.

Present information of fluorescence microscope experiment- name of applied microscope model, manufacturer, country of origin, excitation and emission filters applied.

Table S3- write latin plant names in italics

Typhos;

Line 50, 52- add space before square bracket

Line 149- remove dot before word „and”.

Line 314 write not pymal but PyMOL, correct also in 449

Line 448- write HDOCK

Comments on the Quality of English Language

Minor editing of English language required

Author Response

Response to Reviewer 1 :

We wish to thank reviewer 1 for his/her strong support and valuable comments on this work.

1. Fig. 3

Provide sequences of presented protein motifs, together with their putative function in the additional supplement file or table. 

Reply: Thank you for your valuable comment. Following your suggestion, the sequences of OsSHMT protein motifs, as well as their supposed functions, are provided in new Table S4.

2. Figure 4B- why so much trans-factors are bound by OsSHMT2 (187) and Os SHMT4 (80) ? Could Authors explain this difference based on own observations or references?

Reply: OsSHMT2 and OsSHMT4 stand out with a significant abundance of stress-responsive elements and transcription factor binding sites among the five OsSHMT genes. Notably, within the promoter region of OsSHMT2 and OsSHMT4, there are multiple binding sites for ERF transcription factors (new Table S5). The AP2/ERF transcription factor families are widely distributed in plants and play crucial roles in regulating responses to abiotic stresses, as well as influencing plant growth and development. The promoter region of OsSHMT2 and OsSHMT4 is under the control of multiple ERF transcription factors, which could be involved in various biological processes, including the regulation of specific gene transcription and participation in stress responses. The presence of multiple ERF transcription factor binding sites suggests a complex gene regulatory network, where different transcription factors interact to collectively regulate promoter activity. Therefore, OsSHMT2 and OsSHMT4 likely play significant roles in rice development and its response to environmental stresses.

3. Section 4.6

Provide temperature, length of day and night, intensity of light-if available, during plant cultivation.

Provide name of applied vector. How the DNA fragments were inserted into vector-provide this information, method of cloning, for example applied restriction sites, how DNA fragments were amplified,

Provide details of protoplast preparing and  transformation.

Present information of fluorescence microscope experiment- name of applied microscope model, manufacturer, country of origin, excitation and emission filters applied.

Reply: Thank you for your comment. As you suggested, we have incorporated the environmental parameters relevant to plant cultivation in Section 4.6. Additionally, we have included the names of application vectors, primers, and restriction sites in new Table S9. Furthermore, references for protoplast preparing and transformation have been cited. Detailed information regarding fluorescence microscopy experiments is also provided, as well as the excitation and emission filters utilized (new Line 456-467). 

4.Table S3- write latin plant names in italics

Reply: Thank you for your careful comment. We have written the Latin plant names in italics in new Table S3.

5. Typhos;

Line 50, 52- add space before square bracket

Line 149- remove dot before word „and”.

Line 314 write not pymal but PyMOL, correct also in 449

Line 448- write HDOCK

Reply: Thank you for your careful comment. We have corrected these in the new manuscript (new Line 51, 53, 156, 332, 479, 480).

Reviewer 2 Report

Comments and Suggestions for Authors

The paper performs and extensive analysis of the SHMT gene family in rice. There are similar reports in the literature describing different families, but this provides interesting novel information. The paper may be published if the following points are adressed:

- line 118: described which are the mentioned similar characteristics.

- Fig 1: please, in the legend include the pictogram in parenthesis after the specie to enhance the readibility.

- Line 167: conserved.

- Line 277: developed.

- Figure 5: What is SAM in the first line in pannel A?

- Figure 6: This figure is very weak, as it requires a colocalization to clearly define the subcellular localization of each protein. Please, collocalize with nuclear, chloroplastic and mithocondrial markers to precise the subcellular localization of each paralogue. 

Comments on the Quality of English Language

could be improved

Author Response

Response to Reviewer 2:

We acknowledge the reviewer2’s helpful comments and recommendations, which are valuable for improving our manuscript.

  1. line 118: described which are the mentioned similar characteristics.

Reply: Thank you for your careful comments. As you suggested, we have supplemented the description in our revised manuscript (new Line 119-125).

“The molecular weights of the proteins were also predicted. OsSHMT2 was the smallest protein among the five, with the lowest instability index and the biggest grand average of hydropathicity. All five SHMTs were distributed on different chromosomes but with similar characteristics.” has been changed into “All OsSHMTs, excluding OsSHMT2, displayed an instability index (II) greater than 40, indicating a prevalence of unstable proteins within this gene family. It’s worth noting that all OsSHMTs had a grand average of hydropathicity (GRAVY) less than 0, indicating their hydrophilic properties. Among them, OsSHMT2 was the smallest protein among the five, with the lowest instability index and the biggest grand average of hydropathicity. All five SHMTs were distributed on different chromosomes but with  similar MWs and hydrophilic characteristics. ”

  1. Fig 1: please, in the legend include the pictogram in parenthesis after the specie to enhance the readibility.

Reply: Thank you for your valuable comment. Following your suggestion, we have modified the legend in Figure 1 by adding annotation in parentheses after the species (new Line 143-149).

  1. - Line 167: conserved.

Reply: Thank you for your comments. We have changed this in the new manuscript (new Line 176).

  1. Line 277: developed.

Reply: Thank you for your comments. We have changed this in the new manuscript (new Line 239).

  1. Figure 5: What is SAM in the first line in pannel A?

Reply: Thank you for your careful comments. we have supplemented the abbreviation in our revised manuscript (new Line 257-260). “The samples were obtained from shoot apical meristems (SAM), seedling roots, five stages of developing seeds, six stages of developing panicles, young leaves and mature leaves.” 

  1. Figure 6: This figure is very weak, as it requires a colocalizationto clearly define the subcellular localization of each protein. Please, collocalize with nuclear, chloroplastic and mithocondrial markersto precise the subcellular localization of each paralogue.

Reply: Thank you very much for your careful comments. We have now added supplementary subcellular localization data for OsSHMT2, OsSHMT3, OsSHMT4, and OsSHMT5, along with their corresponding organelle markers, in the new Figure S1. Subcellular localization of rice protoplasts for OsSHMT1 has been previously reported in other articles (Wu, J.; Zhang, Z.; Zhang, Q.; Han, X.; Gu, X.; Lu, T. The Molecular Cloning and Clarification of a Photorespiratory Mutant, Oscdm1, Using Enhancer Trapping. Front Genet 2015, 6, 226.).

Round 2

Reviewer 1 Report

Comments and Suggestions for Authors

Authors significantly improved the manuscript, I have no other comments. Only small typhos could be improved;

Line 485- write: Table S2.

Line 487- write: in selected.

Line 488- write: Original.

Reviewer 2 Report

Comments and Suggestions for Authors

Authors have greatly improved the manuscript.

I can recommend the acceptance.

Comments on the Quality of English Language

Needs some minor improvement